# Irish Equine Industry Stakeholder Perspectives of Objective Technology for Biomechanical Analyses in the Field

**DOI:** 10.3390/ani9080539

**Published:** 2019-08-08

**Authors:** Sonja Egan, Pieter Brama, Denise McGrath

**Affiliations:** 1Institute for Sport and Health, School of Public Health, Physiotherapy and Sports Science, University College Dublin, Dublin, Ireland; 2Section Veterinary Clinical Sciences, School of Veterinary Medicine, University College Dublin, Dublin, Ireland

**Keywords:** equine gait, technology, user-design, subjective analysis, sport horse, thoroughbred

## Abstract

**Simple Summary:**

Technology is becoming increasingly popular across scientific and general population groups. Pedometers, activity and health trackers, are widely used and commercially available. However, these technologies do not appear to be used in field-based equestrian health and performance. Irish industry stakeholders were interviewed regarding their perceived value of technology in equestrian sport. The interviews resulted in four primary themes around horse health, training and management. The movement assessment of the horse is undertaken subjectively through the eye and is incorporated into a holistic management structure that is based on years of experience. There is no such thing as the perfect horse and each one must be treated as an individual. The stakeholders are aware of technologies for movement analysis but demonstrate a healthy scepticism towards new, unproven technologies. Finally, the economic impact of applying technology across the herd is a major barrier to technology use. The results of this study suggest technology design needs to take place in consultation with industry stakeholders to improve field-based use.

**Abstract:**

Wearable sensing technologies are increasingly used in human and equine gait research to improve ecological validity of research findings. It is unclear how these tools have penetrated the equine industry or what perspectives industry stakeholders’ hold in relation to these relatively new devices. Semi-structured interviews were conducted with Irish equine industry stakeholders to understand their perception of objective tools for biomechanical analysis in the field. The study participants came from professional/elite backgrounds in both the sport horse (*n* = 6) and thoroughbred (*n* = 6) sectors. The interview data were analysed using thematic analysis, resulting in four analytical themes. The first theme conveys the importance of tacit knowledge and experience in the holistic analysis of a horse. Theme two highlights that the perfect horse does not exist therefore, equine athlete management is complex and requires a multi-layered problem-solving approach. Theme three describes an awareness among stakeholders of technologies, however they are sceptical of their value. The final theme identified that one of the key barriers to technology adoption is the economic value of the horse and the cost of implementing technology herd-wide. Our findings highlight the need for a user-centred design in this domain, which requires greater consultation and learning between technology developers and equine stakeholders to develop fit-for-purpose analysis and monitoring tools.

## 1. Introduction

Equine gait analysis has been at the forefront of equine research and technological developments since the 1800s. A considerable amount of the literature surrounding equine motion is focused on the use of high-speed motion capture [1] with offline joint angle analyses conducted at a later stage. The research frequently incorporates force plates [2,3,4] and 3-dimensional motion capture tools such as VICON [5] and CODA motion systems. These systems offer highly accurate and reliable data capture opportunities. However, given that equine locomotion in the applied setting is dynamic and takes place over ground, and that the context of an environment influences behavioural parameters [6], the constrained treadmill environment may produce data that do not map very well to the real-world, thus lacking in ecological validity. The data produced is of high quality but is inherently limited in dataset length and appropriate environmental context. Similar issues have been mooted in the field of human sports science for some time where inferences on competitive performance made from laboratory derived data have been challenged [6,7].

The introduction of low-cost, lightweight, wireless, sensor and 2-D video devices has inspired the field-based investigation of potential performance capability with enhanced ecological validity [8]. The implementation of objective gait analyses in the field to assist decision-making by mitigating the inherent subjectivity of human observation-based gait analysis can be facilitated by such technologies [9]. The issues around the subjectivity of gait analysis have been highlighted in the literature where it has been shown that clinical expertise does not improve reliability of lameness assessment [10] and many owners believe their horses are sound despite over 72.5% presenting as the above existing asymmetry thresholds [11]. Improved repeated-measures of equines may also be exploited using objective data-based approaches that would identify subtle changes resulting from injury or treatment that can not be easily perceived by the “eye” [9]. Thus, the theoretical need for objective gait analysis tools in the field seems clear. However, the potential impact of objective gait analysis, demonstrated in many research studies [12,13,14], has not been widely translated to the applied field. There are a small number of commercial equine gait analysis tools available on the market, with varying levels of supporting validity and reliability research evidence. These tools include: EquiMoves—an inertial sensor based system that can accurately and reliably capture equine motion and detect asymmetry [15]; similar inertial sensor based technologies include Equisense and Lameness Locator, the latter of the two being more established with significantly more research to support the validity and reliability of the system [16,17]; Tekscan’s Hoof™ System or Animal Walkway™ System that uses pressure mapping to analyse gait; StrideMASTER that uses GPS for the calculation of stride length, frequency and duration; FotoSelect^®^, a conformational analysis tool that interpolates bone and joint structure based on a conformational photograph and a marker placed on the hip; high speed cameras that facilitate 2-D video analysis using various video analysis software that enable the calculations of stride parameters and joint angles. These tools are promising in terms of the objective information that they can provide to augment a subjective analysis when the principal elements of equine analysis that have been reported in the literature are considered i.e., conformation, asymmetry, power and stride parameters. [14,18,19]. However, these tools either require a certain amount of expertise to analyse and interpret the raw data, or they apply automatic algorithms that produce values that are then ascribed clinical meaning based on a generally defined “normal”.

In the field of technology design and pervasive computing, it is accepted that a theoretical need-as outlined in the previous paragraph for example—and the actual needs and desires of end-users may not in fact overlap [20]. The adoption of technologies is underpinned by sociological and psychological processes that are necessary to acclimatise people to new ways of interacting with information. If the end-users are involved as co-designers in the initial phases of technology development, they are facilitated in learning how these new capabilities can be applied and they can then direct designers towards what they need these new tools to do for them, which can often be outside the designers’ frame of reference [21]. This enables a mutually beneficial journey towards practical innovation. To the authors’ knowledge, no studies currently exist that examine equine industry stakeholder perspectives with respect to technology-enabled, objective equine gait analysis practices and tools.

The aim of this research was thus to understand how existing movement analysis practices are carried out within the elite level of the Irish equine industry. The authors seek to examine the perspectives of a variety of industry stakeholders regarding objective gait analyses tools in the field setting.. Ireland’s dense per capita equine population [22], the economic viability of equines in Ireland and the popularity of Irish native and crossbreeds globally [23,24,25] provides a rich context for this qualitative study. Weary et al. [26] stated that while investigating animal welfare practices, social and cultural context must be considered because purely science-based solutions may not resonate with intended users. This study is applying this same logic to equine biomechanical analysis. It is envisioned that our research will support the development of future fit-for-purpose objective analysis and monitoring systems that can add value in real-world settings.

## 2. Materials and Methods 

### 2.1. Research Question and Methodology

#### 2.1.1. Participants

The participants were purposively sampled for their elite expertise within the equine industry. All participants had international experience at the highest level in their respective discipline e.g., competing, coaching, breeding, treating and/or training. Given the potential differences between the sectors within the equine industry, six individuals from the thoroughbred (TB) (horse racing) sector and six individuals from the sport horse (SH) (eventing, dressage, showjumping, cross-country, etc.) sector were recruited through existing networks, word of mouth, flyers and e-posters. Ethical approval was received from the University College Dublin Human Research Ethics Committee (LS-17-111).

Twelve participants were interviewed with a male to female ratio of 9:3. The Irish thoroughbred stakeholders (TBS) worked with Black type and/or Grade 1 horses in flat and national hunt racing. The interviewees held expertise in more than one area including but not limited to: Elite stud management, bloodstock agents, veterinary medicine, (racing) betting industry, racehorse training and sales. The sport horse stakeholders (SHS) were characterised by Olympic equestrian disciplines and again held expertise in more than one area including but not limited to: 1–5 * international eventing, grand prix showjumping, course design, veterinary medicine, breeding and production. Subject age was not captured under the interview process, rather the subjects were asked to provide an estimate of how many years of applied experience they held in their respective industries (18 ± 8.6 years). The stakeholders excluded their initial childhood years spent competing/owning ponies from this estimation, therefore lifetime horse experience is underestimated by the above value.

#### 2.1.2. Data Collection

In-depth interviews were used to address the aim of this study. An interview guide was designed and piloted, after which one further question was added in relation to the existing analysis processes. The interview was designed based on open questions (a) to understand the thought processes underlying the participant’s existing analysis approach, and (b) to understand their knowledge and perceptions of objective movement analysis tools. A semi-structured approach was employed that allowed the participants to introduce concepts and topics of their own that were then further probed by the interviewer. This ensured that the research findings would be derived from the phenomena described by the participants. Participants were given the choice to complete a face-to-face interview (*n* = 2) or over the phone (*n* = 10). All interviews were conducted by the first author who has experience in qualitative interviewing. The interviews were recorded using a Dictaphone (SONY ICD PX333 Digital Voice Recorder) and lasted between 45 min to 1 h. Once all the interviews were completed, they were transcribed verbatim.

#### 2.1.3. Theoretical Framework

This research is guided by a theoretical framework based on pragmatism. Pragmatism as a framework does not make a stand about truth or reality, rather it posits that whatever promotes inquiry and furthers understanding is good, and whatever restrains it is bad. It understands that knowledge is vulnerable to experience and thus focuses on outcomes in real-world applications rather than on abstract principles [21]. Raitt (1979) stated “We do not ask if it is true, only if it works—we validate not verify” (p. 835), a position that aligns with pragmatism [27]. Since this study is interested in the feasibility of the results for real-world applications, pragmatism underpins our work. In the interest of declaring author bias, we believe that in order to develop a deeper understanding of equine gait, appropriate objective techniques should be introduced to complement and support an experienced subjective assessment. This statement outlines the philosophical position of the authors but is not an isolated idea in the field and has been supported by several research groups [9,12,28,29,30]. In contrast, there have been reasonable concerns expressed in the literature in relation to this position, suggesting that human-eye assessment is superior to objective technologies and that the automation of equine gait analysis is not a desirable goal [31,32]. 

#### 2.1.4. Data Analysis

The transcripts were analysed using thematic analysis [33,34]. This allows the researcher to go beyond a primary summary of the presented data and explore the underlying concepts and meaning of the data set. The primary objective of thematic analysis is to identify patterns in the data. Braun, Clarke and Weate suggested six interviews for meaningful pattern detection, satisfied here under the sample size of 12 [33]. The data analysis was primarily guided by Braun, Clarke and Weate guidelines for thematic analysis, outlined in six phases [33]. 

Phase 1–2: The interview data was transcribed verbatim following the interviews and cross referenced with notes taken during the interview process to provide additional data depth. Each transcription was then read numerous times to construct context of the information prior to beginning the analysis process. The codes or labels were applied to the data with specific relevance to the research question or data patterning. One interview was first coded separately by two researchers and then compared. In-depth discussions took place between both researchers in relation to the assigned codes, particularly in relation to the semantic and latent interpretations of the data and reflections on researcher bias. Two further interviews were coded separately by two researchers and then compared and discussed. The first author then proceeded to code the remaining interviews. The second researcher (DMcG) then reviewed the codes against the transcripts, challenging and/or adding codes where appropriate. The codes were then combined across the data set by inputting codes into an Excel spreadsheet. 

Phase 3–5: The codes were developed into several descriptive themes by the first author and then reviewed with the second researcher. Both researchers then worked together iteratively to collapse the descriptive themes. Using clustering and visualisation techniques and checking back on the raw data, analytical themes were developed that faithfully represented the raw data and addressed the research question. Finally, the four overarching themes were named and defined. 

Phase 6: The results and discussion were written iteratively to critically develop the analytical narrative during the analysis process. Important quotes were extracted from the data to convey the concepts arising from analytical themes. 

Determining the quality of the thematic analysis is an important issue and while Braun and Clarke [35] do not typically recommend a codebook, they do suggest notes made throughout the coding and analytical processes. To this end the authors kept a hybrid of the two concepts, allowing familiarity and constant reflection of previously coded data in the bespoke Excel sheet. The documentation of the decision-making process was updated regularly to ensure consistency and transparency of the data analysis. This also facilitated constant comparison to the existing codes or how they were developing within and between interview transcripts. Additionally, the 15-point thematic analysis checklist was applied to ensure the quality and comprehensive nature of the analysis [35]. The thematic analysis implemented was also guided by additional methodological research by Harden (2008) and Ward et al. 2009 [35,36,37]. 

## 3. Results and Discussion

Thematic analysis of the transcripts produced four major analytical themes that are presented and discussed in this section.

**Theme** **1.***Biomechanical information is elicited through subjective analysis and integrated into an overall whole, based on holistic analysis and tacit knowledge*. 

The participants consistently acknowledged that existing analyses processes are subjective, as echoed in the literature [38]. However, the stakeholders frequently refer to their tacit knowledge and experience as the most important aspect of identifying good horses and maintaining high-quality animal management structures.
“Sometimes a lot of it might be tacit knowledge whereby you look, and it doesn’t look quite right and you don’t even know exactly why. And then other times you look and go, wow that is lovely today. Again, it is hard to pinpoint exactly why but it is kind of the overall balance, it looks like it is coming together better”.(SHS, 2)
When describing certain aspects of their own analysis such as monitoring routine training, participants consistently referred to “feeling” when a horse was progressing or not moving normally. When they were pushed to define what this feeling was or how it is visually represented, they struggled to explain it and stated, “you just know”. This knowing is highly contextual and holistic where participants trust their experience, horsemanship and husbandry skills to tell them if the horse is appropriately prepared for the next level of training or competition:
“you would know from the feel of them, if the feel is the same or not, and again that comes down to the black art of horsemanship. But apart from what makes a good rider is to be able to know when their horse feels good or not”.(SHS, 2)

Tacit knowledge is defined as the inarticulate knowledge which cannot be explicitly taught but rather acquired through direct or on-the-job experience [39]. It has also been described as the knowing how rather than knowing what [40]. This type of knowledge is particularly relevant to equestrianism given the nature of working with nonverbal yet expressive animals. This was depicted vividly in a paper by Butler (2017) who speaks about “the feel for the game”, specifically in relation to the mastery of an equine racing yard. The author describes the “feel for the game” as “an intuitive bodily awareness and attitude that occurs through familiarity and repetition, a process of both doing and observing empathetically, of reading bodily nuances, nonverbal cues, imperceptible movements, a fleeting expression on [a] face”, and indeed, in the face of a horse [41].

The interviewees frequently referred to the generational knowledge developed through working with horses, often in familial networks from a young age. This experience has shaped their ideas and beliefs around determining optimal equine performance and management structures.
“Probably what your father and mother did at the time. You pick up a lot from them at an early age”.(SHS, 1)
Nash and Collins (2006) describe how coaches rely on personal contextual knowledge to inform decision making. The decisions are generated from an immediate understanding of a situation, disregarding the need for deductive thinking. This may explain the seemingly instinctive decision making associated with tacit knowledge [42]. 

Our data suggests that early entry into the equine world and working with highly experienced individuals over many years enables industry stakeholders to unconsciously access previous experience to immediately inform present decision-making. This is particularly important for TBS and the restrictive nature of thoroughbred purposed auction and sales. This context has driven the need for a skilled and experienced eye to inspect correct conformation and “the big walk” when trying to discern a superior performer. Those working in the thoroughbred industry noted that it is difficult to obtain a clear and timely inspection of horses in the sales setting. This gives rise to certain tricks, such as shoe fillers, that can be used by sellers to mask a movement or conformational abnormality (toed-in/out, knee valgus/varus), suggesting that it is difficult to believe what you see in the sales ring. They outlined that in an ideal world, one would like to see the horse gallop but noted that this is not feasible under current practice, and often they rely on their imagination and the mind’s eye to predict how the horse may move on the gallops:
“you like to use [your] imagination when your look at the sales because you can’t gallop them at that stage but that’s essentially what you’re looking for, that lovely easy pendulum where the toe flicks out in front and they place the foot down”.(TBS, 6)
The interviews did capture however, that thoroughbred sales infrastructure is adapting due to the availability of technology, driven by the commercial impact of detecting a valuable racehorse. This exists primarily in two forms:
Breeze-Up sales for flat racehorses, originating in America but are becoming increasingly common in Europe. Two-year-old horses that have not yet raced are galloped (breezed) up over a short distance (two–three furlongs) and timed in an attempt to predict their performance potential. This allows buyers to see the horse move at the gallop which has not been a traditional component of Irish thoroughbred sales. On certain auction tracks, independent companies offer exclusive video analysis packages, producing a report based on frame by frame video inspection of a galloping horse. The first and only Irish Breeze-Up sale of flat horses was conducted in 2019.A form of 2D video analysis using a marker set up which records the horses in walk outside the sales ring and is analysed offsite. The company provides a report to the interested party theorising the horse’s potential ability based on a breakdown of segment lengths, joint angles, etc.
Despite this, TBS highlighted that it was “uncanny” the number of good trainers who still manage to pick up the good horses at the breeze-up sales—regardless of tools available. This tacit knowing and holistic expert eye highlights the challenge in establishing reductive thresholds using objective tools that could potentially categorise horses as normal/abnormal or good/bad.

**Theme** **2.***There is no such thing as perfect conformation or movement in the real world; management of injury risk and performance is a complex system requiring multi-layered problem-solving approaches*.

The interviewees described key features of movement and animal phenotype that contribute to the optimal or classic horse that theoretically will lead to greater performance potential. However, they also consistently stated that there is no such thing as the classic horse in practice and everything is managed on an individual basis. When asked if they were aware of any horse who performed well with unconventional conformation or movement characteristics, all participants answered easily, and no two answers were the same. They frequently referred to these horses as “freaks” but also pointed out that this phenomenon was common, highlighting the individuality of horses. It can be suggested that conformation exists on a spectrum and it is only abnormalities lying on the extremes that would drastically impair a horse’s ability. Conformation was defined as a key cornerstone of a horse’s potential, ultimately impairing the horses career longevity depending on wear and tear issues.
“If they have got crooked limbs, they may not preclude a good effort but over time they don’t have the same longevity because of strains in certain areas. It just gets referred and they don’t last over time”.(TBS, 1)
“yea I mean they come in every shape and size […] generally, the good horses that last have the correct conformation, but there’s always the exception to the rule”.(SHS, 6)
The participants explained that there were certain faults or abnormalities they would accept or reject when purchasing a horse. These acceptable abnormalities were not uniform across the cohort and were largely driven by the horse being analysed, budget and previous positive or negative experiences.
“like the really big buyer, that’s what they’ll look for straight away; they won’t forgive conformation faults because they’ll be buying for other clients. But, clients who want racehorses and are trying to look for a bargain will maybe forgive a horse that’s toed-in or has a slight issue with an x-ray or something like that as long as they have that walk”.(TBS, 6)
The stakeholders believed an established horse’s performance record was the most important indicator of ability and would supersede any abnormality as the horse had proven its performance capability. Additionally, a horse’s temperament and attitude was frequently referred to as an important indicator to determine the wellbeing of the horse, stating that if the horse is ”happy” or ”enjoying his work”, this was the best indicator of overall health and performance. 

Managing pain in horses was an especially complex issue, requiring deep knowledge of individual horses and knowing when to rest, reduce work or when to enlist help from other professionals. They stated that they expect performance horses to have “pain” or “soreness” or “niggles”, and they relied on experience and intuition to determine if the pain was normal muscle soreness or an issue requiring intervention. These terms were used interchangeably to convey an issue however often one term was used to define presentation of heightened severity, which differed across interviewees.
“most horses in competition will have niggles and pains and stiffness even if it is not soreness so optimal is a fluid open step, not displaying any signs of soreness or stiffness, I mean that is optimal”.(SHS, 2)

It is evident that the study participants’ priority was to maintain the horse pain-free for both welfare and performance reasons. However, they explained that it can be difficult to get to the bottom of a problem when the presenting condition is originating from a subtle issue elsewhere. As part of a multi-layered problem-solving approach, these industry stakeholders clearly value engaging in multidisciplinary teams to address painful conditions, however frustration was expressed around honest referrals and regulation. They describe how horse owners pay anything to “get them right” but there is no clear path to organising a meaningful investigation between specialists (e.g., physiotherapist, vet, farrier etc.), and owners can be hard done by.
“That there are no known standards and quantities that we can actually refer to and it all becomes very cloak and dagger”.(SHS, 2)
“I think therapists of all types, first they need to be regulated so people are getting in what they actually pay for and so that people are trained and educated in the service that they’re offering—they should all be able to work together to give the client and the horse the best holistic treatment.”.(SHS, 4)

In addition to veterinarians and manual therapists, the participants consistently highlighted the importance of the farrier’s role in correcting conformational faults and maintaining the horse’s healthy movement. The individualistic approach to managing each horse was a common theme across interviews:
“There are five horses, or ten horses and they have all different conformation and different stride patterns and we will train them. You can’t have placebos in this, you can’t have standards because there are so many complicated [aspects], there is genetics...and blah blah blah”.(TBS, 4)
This was also reflected in the shoeing approaches that varied widely depending on the horse’s needs, the management team’s preferences, the discipline and the economic viability. There was a great deal of trust placed in the farrier’s ability to choose the right shoeing option for the horse that could be corrective and/or protective:
“I’ve a very, very good farrier and I would discuss options with him, but I will always take his advice because it’s his area… and I would discuss it with him because if you’ve a good farrier who knows what he’s doing, their advice is good enough for me.”(SHS, 4)
In the TB stud environment, farrier intervention was deemed critical in the young foal showing moderate to severe conformation abnormalities. The stakeholders working in these environments outlined that they were slow to intervene with aggressive clinical interventions and would typically begin with the farrier gradually rasping or applying plastic corrective shoes to foals to promote optimal skeletal development. 

Jönsson et al. (2014) found a positive correlation of 4–5 year old conformation, overall health and performance to career longevity in Swedish warmblood horses [43]. This relationship between conformation was also evident in our data. Collaborative working relationships between vets, farriers and owners in discussing shoeing options and solutions to foot problems, based on objective screenings where necessary, have been shown to be effective in maintaining horse soundness and hoof quality [44,45]. These previous research findings appear to be borne out in our data where collaboration with vets and farriers were considered vital aspects of maintaining a functional, healthy horse. Research investigating racehorse welfare has shown that the one size fits all approach indicated the minimum welfare standards were met, whereas horses perceived to have better lives were trained and treated as individuals [34]. It is clear from our data that for better health and thus better performance outcomes, horses should be treated as individuals. 

**Theme** **3.***Stakeholders have an awareness of technologies that enable objective movement analysis and acknowledge the competitive edge that they may offer, yet scepticism persists*. 

Many stakeholders had experience working with a movement specialist who they perceived to be an expert in the field or a person using a form of technology.
“I’d consider that the likes of your great trainers, your [name XX] who pick out horses at horse sales. When you look at the breeze up sales, I’d consider all them movement analysis specialists”.(TBS, 6)
They were knowledgeable in relation to existing technologies for objective movement analysis. They expressed interest and sometimes idealised the use of technologies and how it could help to get additional small percentages of performance improvements which could mean the difference between winning and runner up. Most of the participants referred to the need for marginal gains in such an elite, competitive setting and they recognised that the use of technology may have the potential to deliver this:
“I think 90% of horses will perform to the same ability no matter what trainer is training them once they’re fit, but it’s that extra 10% is the reason that the great ones become great. Each little % or half % that you could get up the ladder by having technology or you know by having the best jockey on board […] next 10 trainers down the list could have spotted that as well and therefore that half a % difference between the really really good ones and the rest comes a little bit closer”.(TBS, 6)
“the difference in coming 9th or winning, you know people often think that O give up on a horse that maybe keeps coming outside the top 6 and decide the horse isn’t that good. But it might be something very small in terms of the training, of way of going, and I suppose those kinds of technologies might help people figure that out”.(SHS, 5)

The TBS often described their experience of breeze-up sales abroad and how technology was being used to select the horses with the best stride characteristics. This information generated from 2-dimensional video analysis was apparently sold exclusively to a small number of potential buyers. This provided these individuals with augmented information other buyers could not access, thus affording them the opportunity to gain an edge in discerning the horses with the best potential ability based on stride mechanics. Additionally, lameness and objective assessment of lameness was consistently mentioned across all interviews despite the fact the authors never mentioned lameness in the recruitment information leaflet or interview questioning. This is perhaps not surprising given that clinical lameness is reported as one of the most prominent reasons for wastage, the time lost to training and gait abnormality in horses, as evidenced in several studies [46,47,48]. Technologies based on wearable inertial sensor units (accelerometers and/or gyroscopes) have been commercialised to assess lameness and the participants were aware of this development, although not always persuaded of its value. While it was acknowledged that technology could potentially provide a competitive edge, it was recognised that there are many other under explored aspects of performance that can be leveraged, beyond technology, for example elevating the rider’s state of preparation:
“if you want to be at a more competitive level you’ll look at ways and means to help you achieve that, so if you look over at the last 10–15 years even in Sport Ireland […] they’re focusing [on] psychology, …mentally correct general fitness of the athlete as well as the horse so it’s a ... whole picture to performance and that’s really because other nations are doing it and succeeding. One thing you can’t take away from the Irish is the raw talent; one thing you don’t want them to focus on is the raw talent, you want them to put it all together”.(SHS, 1)

Despite their awareness of existing technologies, few employed them frequently or were informed of how they actually worked. This was related to a degree of uncertainty and scepticism surrounding how a device captures the data, i.e., something is done, and the results are produced. They questioned this black box scenario, where they were dubious about the accuracy and reliability of the results stemming from a process that is essentially hidden and sometimes lacking supporting evidence. They additionally expressed concern over the tools that may be based on pseudo-science, lacking any real therapeutic or performance benefits, citing the lack of regulation as a contentious issue.
“I think we need more consistency and we need regulation of people using pseudo-science and managing to trick people out of their money with ridiculous things […] yea I get very frustrated with the pseudo-science... magic boxes that you put on your horses in this box […] shine a light on them give you a read out on what’s wrong with them, shine the light on them for another hour and its fixed”.(SHS, 4)

Many of the participants somewhat apprehensively acknowledged that technology will become more prevalent in the industry, stating that those who do not move with the times will find it difficult to progress, but as it stands, technology has not been the fail-safe method of finding the ideal horse, and they doubted that it ever would be. They spoke about the naïve misuse of the data produced by these technologies suggesting that it would be quite easy for inexperienced individuals to use these tools blindly in the absence of basic horse husbandry.
“could be a case of putting the cart before the horse very much because there’s such a lack of understanding of biomechanics in the industry, and better horse husbandry and horsemanship, introducing that technology would really bypass […] some of the natural skills that people need”.(SHS, 5)
Further to what one might call a healthy scepticism of science and technology, some fears were expressed by the participants in relation to the future of technologies in the field, such as how it may impact their livelihood and the horses’ economic viability.
“It hasn’t fully caught on like I would have expected it to, and we seem to be slow to embrace technology and we are almost fearful of what technology can do. It is like people from a job security point of view, they worry is it going to mean if people know which the good horses are does it mean the average and the bad horses, we won’t get paid for them in the sales?”(TBS, 2)
It was also suggested that the traditional methods of older generations may be a prominent reason for the slow uptake of technology in equine industries. Participants discussed the fact that younger trainers and riders may be more likely to try technologies as they are more familiar with their use and are willing to explore these avenues in order to enhance their ability to train or ride. Older equestrians on the other hand sometimes have the attitude that if they cannot see or identify optimal or abnormal themselves, then they should not be working with horses:
“the heart rate monitor I think is interesting but again I think it will only show me stuff that I should already know”.(TBS, 6)

The technology generation-gap is a known phenomenon [49]. Tacken et al. (2005) stated that there was a lack of motivation in those aged 55 years and older to engage with technology due to the difficulty perceiving and navigating the technology interface [50]. Dickison et al. (2007) reported usability issues as a barrier to adoption, indicating that older adults may not perceive the information as relevant to them [51]. Lack of knowledge of technological devices and operability is also highlighted in the literature as a major barrier to adoption [52,53]. Customisation is one of the largest end-user issues with technologies, citing that many interfaces are locked in, prohibiting the user from making meaningful changes and updates to the functionality of an interface [54]. It appears that there is a research-practice divide in the equine performance setting. Indeed, this was evident from our interview data. Participants acknowledged the value of research and qualifications to engender trust but feel a disjoint between certain research investigations versus what the applied professional or industry stakeholder requires, losing the common-sense aspect of the application and engendering scepticism. They suggested that they would only be convinced by new tools or technologies if a reputable individual they knew endorsed their value. Similarly, stakeholders described how they would only use expertise within their circle for fear that the quality of work done by another individual would be sub-par. This is suggestive of a high level of discernment, which is perhaps to be expected from stakeholders working in the elite field. Lack of regulation has been identified as an ongoing issue in equine nutritional supplementation [55,56,57] and stakeholders may fear similar is happening with technologies that supposedly can enhance equine analysis and performance.

**Theme** **4.***The economic value of the horse and the cost of the tools available is a primary consideration for stakeholders*.

The most prominent barrier preventing stakeholders from applying the existing tools and technologies was the cost of the tool and the value of the animal at hand. The horse’s value extended right across into methods of management, with owners suggesting they would change shoes or use these tools if the animal’s worth warranted it.
“I think you couldn’t afford to pay it! ... If I had an Olympic superstar, I’d think about it”.(SHS, 1)
An example of financial pressure was anecdotally explained by one of the participants in our study, citing another trainer who had written that €1 a day may sound very little, but many small-time trainers and equestrians cannot justify that cost when it is applied across their yard on a consistent basis.
“margins are so tight; you see the amount of trainers who have gone out of business because they are making a loss. They can’t add an extra €1 or €2 a day for each horse which will add up to whatever, a couple of thousand a year when they’re already not making anything”.(TBS, 6)
In the elite sector, the horse is viewed an as investment and the use of technology can be viewed as follows: it may benefit the horse and its performance, therefore increase the sales profit margin or it could have no tangible impact on management practice and therefore cost the stakeholder more to maintain and run the system.
“horses from our point of view are an investment and the more analysis and more information we have to help us make better decisions the better decisions we will make, and the better decisions we will make will have a more positive effect on how our investments prosper going forward or how we manage them”.(TBS, 2)

Stable size was put forward as a possible driver for adopting objective monitoring tools however there were opposing views on this within our cohort:Larger organisations can afford to use technologies as they have a larger cohort of horses with greater time and financial support which could make use of novel technologies, as highlighted by this quotation:
“But I think if you’re the likes of XX or XX or whatever huge numbers [...] if you were looking at it you could say well every horse must walk through this particular office every morning and trot and it’s recorded on a machine that everyone can look back on so it’s part of a management program”.(SHS, 1)Trialling technologies may be easier in a small yard with less horses, with reduced implementation costs and time investment to collect and interpret each horse’s data.
“If somebody has two yearlings at home and wants to try out a new technology, fine, but if we are going to try it out we have to do it for the whole herd and whatever that cost is you are multiplying probably by 750, so you would want to be pretty convinced it is going to work”.(TBS, 1)

Clearly, it is difficult for stakeholders to envision how these tools would fit in an applied stable setting in a way that is economically viable. Some participants expressed frustration at a new type of pressure in recent years, where consumers are no long willing to wait for a horse to mature and develop, inhibiting their ability to reach athletic maturation. This has influenced breeding and management practices, effecting the horses’ career longevity. For example, a horse may be engaged in early training and potential overloading to the detriment of its development. This approach may reap short term profit but could also impact the horses’ longevity in a case of too much too soon. The stakeholders believe that this is driven by results, the betting industry and overall consumer impatience. Could this be coincidentally driven by wide-spread consumer use of smart technologies and availability of entertainment and services on-demand? It is worth considering how the availability of objective movement data might influence this trend.
“I think times are changing big time, society wise and people want.. they don’t want to be waiting around for horses to run anymore. If your hobby is racehorses, they want their horses running now, they don’t want to wait for them until [they are] three years old”.(TBS, 2)

Thus, there is also an “opportunity cost” involved in the application of these technologies, given the pressurised environment described above. If there is time being invested in technology related workflows, there is time being taken away from somewhere else. The focus on the cost implication of using objective technologies evident in our data is also captured in a recent study investigating the application of sports science to training and management practice in the racing industry; stakeholders felt they did not have expensive enough horses to justify the investment in sports science [58].

### Adoption of Technology

The Technology Acceptance Model is the most widely applied model of technology use and acceptance [59]. It has been applied to investigate the intentions for technology use across areas such as general information technology, education, sport and health care [60,61,62]. The model theorises that the two main criteria—the perceived ease of use and the perceived usefulness of a technology—predicts behaviour intention and use. Ease of use pertains to the effortless usability of the interface while usefulness describes the ability of the technology to enhance productivity [59]. Tao et al., 2009 found that when the users had little experience of the technology, the focus should be ease of use, rather than usefulness. The intention is to allow the user to familiarise themselves with the technology before increasing system complexity. The user can decide at that stage if they would like to continue to implement the technology or re-evaluate their requirements. The authors state that the user’s behaviour and beliefs change over time and suggest the use of an evaluation system that may help to detect when additional training or system adaptation is required [60]. Venkatesh et al. (2000) outlined that successful technology integration can enhance productivity but system failure can also lead to economic losses and overall dissatisfaction [59]. The interviewees echoed strongly similar concerns around economic losses and potential overall dissatisfaction. They consistently looped back to the cost-benefit aspect of buying and applying these tools. The stakeholders from the elite equine performance setting who participated in this study are already highly competent and successful in their respective fields. Their expectations of technology solutions go above and beyond their existing equine management structures and must have a tangible impact to justify implementation. They contemplated the value of sophisticated longitudinal datasets but maintained caution around how the data would be interpreted. Who would interpret it? How valid, sensitive and accurate would the data be? Their expectation of a valuable technological support to their workflows thus appears to be a simple and cost-effective device that can integrate large multi-modal data sets, outputting intelligent, reliable and individualised information that enables a holistic view of a specific horse. This information would need to add considerable value to the existing knowledge base—both tacit and overt—contained within the professional team that supports the horse.
“It is not that I wouldn’t use them, I would use a lot of these at jobs and training except they are just not cost efficient either time and motion, the time it takes to use them or the money it takes to purchase and run them. And they don’t make a significant quantum leap to my training regime for me to do it”.(TBS, 4)
From the stakeholders’ perspective, it appears that in their current form, technology applications simply have not crossed their value threshold for widespread, daily use.

## 4. Conclusions

Our data highlighted tacit knowledge as an extremely important factor guiding analysis, monitoring and decision-making surrounding equine health and performance. Artificial intelligence research in business has investigated methods of converting implicit or tacit knowledge to explicit knowledge by using a knowledge exchange and extraction techniques. This facilitates the application of data mining approaches to explore trends and relationships with the potential to uncover new information [63]. From an equine perspective, these approaches could be used to develop data analytics infrastructures where longitudinal data is mined to uncover trends around injury risk and peak performance to support data-driven decision making. However, reaching this level of technology infrastructure and intelligence requires a deep understanding of stakeholders’ practices and culture. Future work should consider how best to engage industry stakeholders in technology design. Ethnographic observation of subjective information gathering, processing and decision-making in the day-to-day management of performance horses would provide a rich research basis on “the black art of horsemanship”, from which useful technologies and data pipelines could be designed. However, as with all elite athlete settings, such access and enquiry may not be realistic, desired or appropriate.

This research probed the elite Irish equine industry stakeholders’ perceptions of objective technologies that capture locomotion data from equine athletes. This maps the cultural attitude regarding the use of technology in the Irish equine industry and therefore may not be directly generalisable across equine industries. A further limitation of this research was the lack of specific dressage perspectives included. There were elite dressage riders who initially indicated interest in being part of the study but were subsequently lost on follow-up.

In their current format, technology solutions—particularly the more “hi-tech” tools as opposed to 2D-video analysis—do not appear to address stakeholder needs. Trusting in the data produced by sensor devices and being able to interpret it correctly were concerns that were raised across the cohort. Movement analyses are valued as an important aspect of maintaining overall equine health and performance, but participants were conscious that movement analyses are one small aspect of a larger functional unit. These stakeholders felt that there is a disjoint between research, commercialisation and equine industry driven objectives. Working from the industry-out rather than the outside-in may provide researchers and equine stakeholders alike with a degree of ownership over technology development and would enable the creation of user-friendly, meaningful feedback interfaces. Current technologies do not appear to provide a suitable basis to holistically judge the progress of their horses beyond the existing knowledge base available within the multi-professional management structures, and as such, are not deemed worthy of investment.

The participants foresee that technology will become increasingly important in supporting methods of equine management in the future. Many interviewees have translated sports science-based training principles, such as evolving their management programs, training methods, using video footage, photos and note-taking to monitor equine movement and development over time. The appetite for continual improvement is clear. Their extensive experience in elite equine sport, often across several fields of expertise and contexts per participant, has provided a rich picture of the existing workflows and perceptions of objective technologies in a variety of elite equestrian settings.

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
