# Peer review of "Irish Equine Industry Stakeholder Perspectives of Objective Technology for Biomechanical Analyses in the Field"

_animals, 2019, doi:10.3390/ani9080539_

Round 1

Reviewer 1 Report

Thank you for the opportunity to read this interesting and novel research paper. The paper was generally very well written and easy to follow. I have some suggestions for re-ordering the introduction and analysis and separating out the discussion section.

The first point I would like to make is that the introduction does not clearly describe the sorts of technologies available, and their relevance and suitability for these industry end users. Without this information, it is quite difficult to evaluate the rest of the paper, including the suitability of the participants recruited, the claims these participants make, and the conclusions that the authors draw.

I suggest you include a clear and detailed paragraph of the types of technologies available to these specific equine professionals (sport horse and thoroughbred) and their reliability and utility. As a reader, I needed to know what end user technologies there are and how useful they are for their intended purpose from a scientific perspective. What do the technologies purport to do for these particular users? The description of the technologies in lines 238-250 seems to me to belong in the introduction. At present, the paragraph (lines 53-62) describes the potential of these technologies, but the reader does not get a clear sense of what advantages they offer over subjective assessment. Do the references 9, 10, and 11 refer to light-weight ambulatory devices, or do these sentences refer to the ability of complex gait analysis technologies to do better than subjective assessments? It is not clear as the following sentences describe the validity, reliability and utility of these devices as uncertain. If the value of these technologies is unclear, then do the lightweight, wireless ambulatory devices offer any reliable advantage over the subjective assessment?

Starting with this technological and scientific perspective will then enable the reader to evaluate whether theoretical need is divorced from the end user experience (lines 63-73).

Similarly, the introduction lacks any review of information on how equine professionals currently evaluate movement in equines. Some of this information is on page 266-269 on the correlation between confirmation and performance. It seems to me that if the aim of the research was to understand how existing movement analysis practices are carried out, then it would be useful to summarise any existing literature on this at the beginning of the paper as part of the introduction rather than dropping it into the analysis. Lines 357-358 also seem to belong in the introduction to give a framework for the paper.

Participants

The description of the sample made me wonder if they would all equally have use for these types of devices. For example, would a course designer have the same type of need to a bloodstock agent? To what degree would these interviewees possess a similar enough view to make thematic analysis suitable? Not sure what an equine entrepreneur is? Is there a more specific term for this?

Theoretical Framework

The theoretical framework seems an unusual choice for this paper. The choice of pragmatism is justified through links with health and education research, rather than through any specific characteristics of this research question. The description is general, and then shifts to a very specific position statement. The authors’ position is that objective techniques should be introduced to support subjective assessment. The rationale for this position (the superiority of objective techniques) was not convincingly established in the introduction. It seems unusual to have this explicitly identified as a driver of the analysis in the theoretical framework.

Good details of the method and coding.

Results and Discussion

Theme one and two seem like analytic themes, and make sense to me. The extracts seem well-chosen and sensible. Across the analysis as a whole, some of the material seems to be discussed in the wrong place. The material on lines 407-414 seems to belong to theme one about the holistic approach. Lines 423-433 and lines 440-442 seems to belong with theme one on the tacit knowledge, rather than theme three on scepticism.

In theme two, the material on using farriers and therapists seems to wander away from the key aspect that there is no perfect confirmation. This theme could be edited down to focus on this core aspect. Lines 417-432 seem to be part of the analysis of theme one on the importance and value of subjective assessments and concerns this value would be lost. Using a table to organise the data might make it easier to see these similar instances across the analysis as a whole.

Theme three seems like a response to a question the interviewers asked rather than a theme arising from the data analysis. In addition, theme three seems to cover three quite separate aspects: awareness of technology, competitive edge, and scepticism. It is not clear to me why these have been grouped together. It might clarify the analysis to separate these out.  I would discuss the awareness of technologies first, and then discuss the participants support for the competitive edge they offer separate from their scepticism.

Theme 4 overlaps with the earlier themes, with the suggestion that technology is not able to provide anything over and above a good eye.

The authors make some odd comparisons to the wider literature in the results and discussion. For example, key performance indicators in business growth and success in line 188, clinical domains of bedside care line 208. It feels like this is done to justify the analysis rather than to genuinely extend the readers understanding of this material.

I suggest the authors remove all the references to the literature from the results and have a separate discussion section. This would situate the whole analysis in a relevant literature (which is probably about end user use of technologies in sport coaching such as the Nash and Collins (2006) reference). This would remove the temptation is situate every minor finding in the literature. At times, the literature comes before the analysis that it is supposed to support, which makes it unclear whether the analysis is driving the literature or the other way around. The material on lines 501-526 would also be better in a stand alone discussion section.

Ethnography is a good suggestion for future research. The limitations section should consider what specific difference this particular participant group had on the findings.

End with a strong statement of the research contribution, rather than ending with the limitations or future research.

Author Response

Dear Reviewer 1,

Thank you for thoughtful and constructive comments. Authors response has been attached below.

Reviewer 2 Report

I enjoyed reading your paper and I thought your quotes gave a real flavour of the perceptions of both racing and wider equine stakeholders.

I only had a couple of points I think would help give more clarity to the paper.

line 48.  You first use the term ecological validity. A line or two to explain what this means would help readers of you paper to understand why this is significant.

138.  Some references made to other studies that have used thematic analysis.

198. 'the feel'. Needs some references to help contextualise what you are referring to. See Butler, D, 2017. Regaining a 'feel for the game' through interspecies sport. Sociology of Sport Journal, 34 (2): 124-135. 

242.  Readers may not understand what 'breeze-ups' are. A brief clarification would help here.

Author Response

Dear Reviewer 2, 

Thank you for you encouraging and thoughtful commments. Please see the attachment for authors reponse.

Round 2

Reviewer 1 Report

The authors have done an excellent job of incorporating the reviewers suggestions. The paper now reads well.